# Statistical Dataset and Data Acquisition System for Monitoring the Voltage and Frequency of the Electrical Network in an Environment Based on *Python* and *Grafana*

Javier Fernández-Morales, Juan-José González-de-la Rosa *, José-María Sierra-Fernández,
Manuel-Jesús Espinosa-Gavira, Olivia Florencias-Oliveros, Agustín Agüera-Pérez,
José-Carlos Palomares-Salas and Paula Remigio-Carmona

Research Group PAIDI-TIC-168, Department of Automation Engineering, Electronics, Architecture and
Computers Networks, Higher-Polytechnic School of Algeciras, University of Cádiz, E-11202 Algeciras, Spain;
javier.fernandezmorales@uca.es (J.F.-M.); josemaria.sierra@uca.es (J.-M.S.-F.);
manuel.espinosa@uca.es (M.-J.E.-G.); olivia.florencias@uca.es (O.F.-O.); agustin.aguera@uca.es (A.A.-P.);
josecarlos.palomares@uca.es (J.-C.P.-S.); paula.remigio@uca.es (P.R.-C.)
* Correspondence: juanjose.delarosa@uca.es

**Abstract:** This article presents a unique dataset, from a public building, of voltage data, acquired using a hybrid measurement solution that combines *Python*$^{TM}$ for acquisition and *Grafana*$^{TM}$ for results representation. This study aims to benefit communities, by demonstrating how to achieve more efficient energy management. The study outlines how to obtain a more realistic vision of the quality of the supply, that is oriented to the monitoring of the state of the network; this should allow for better understanding, which should in turn enable the optimization of the operation and maintenance of power systems. Our work focused on frequency and higher order statistical estimators which, combined with exploratory data analysis techniques, improved the characterization of the shape of the stress signal. These techniques and data, together with the acquisition and monitoring system, present a unique combination of low-cost measurement solutions, which have the underlying benefit of contributing to industrial benchmarking. Our study proposes an effective and versatile system, which can do acquisition, statistical analysis, database management and results representation in less than a second. The system offers a wide variety of graphs to present the results of the analysis, so that the user can observe them and identify, with relative ease, any anomalies in the supply which could damage the sensitive equipment of the correspondent installation. It is a system, therefore, that not only provides information about the power quality, but also significantly contributes to the safety and maintenance of the installation. This system can be practically realized, subject to the availability of internet access.

**Dataset:** https://doi.org/10.7910/DVN/EGI7X1

**Dataset License:** CC0 1.0

**Keywords:** grid frequency; *Grafana*$^{TM}$; higher-order statistics; LabVIEW$^{TM}$; network-attached storage; power quality; *Python*$^{TM}$; statistical signal processing; voltage monitoring

## 1. Introduction

Nowadays, in the context of a modern and smarter electrical grid, there are numerous distributed resources which, although a priori independent, exert mutual influence, causing degradation of the supplied voltage.

Distributed non-linear loads and intermittent energy sources in the electrical network have resulted in a complex body of network state power quality (PQ) data; it is vital to interpret this data correctly, in order to make the necessary compensation, and to

forecast not only demand but also possible network state degradations which obey seasonal behavior, or which could be triggered by unexpected causes. In fact, building a resilient national power system with real-time state monitoring has become a primary goal of many governments [1].

Microgrids gather DG systems, like inverters in photovoltaic panels, wind turbines, chargers of electrical vehicles, etc., which have direct consequences on the power line and in the subsequent quality [2,3]. However, despite the complications that cause this kind of load, the advantages of energy saving, control of energy flow and direction, etc., exceed the possible disadvantages [4].

It is not only new non-linear loads, however, which cause alterations to the network. Traditional non-linear loads, such as transformers and rotating machines, also contribute to network degradation [5].

Other types of network disturbance, such as rapid voltage changes [6] or the effect of lightning strikes [7], could also be detected by the system proposed in this study. While they do not typically receive much attention from power quality standards monitoring, compared to other phenomena, they still have an effect, the detection of which could contribute to an even more complete diagnosis of the network.

Maintaining satisfactory quality is the joint responsibility of the producer, the supplier and the user, in that order of influence, abilities and responsibilities [8].

Recording power quality data over a long period enables tracking of the power consumption of devices and systems. Voltage and frequency are the two basic magnitudes that characterize the quality of the power supplied to consumers, and the operation regime of a power system [9]. The parameters measured by current instrumentation consist basically of amplitude and shape, as an alteration in the latter leads to a variation in the former.

A complex and highly dynamic electricity sector brings to light a twofold perspective. On the one hand, expensive analyzers are used in the industry, which are connected occasionally in seasonal measurement campaigns. On the other hand, there is a growing tendency to develop domestic energy quality indicators that help the novice user to analyse their supply. In both scenarios, the underlying idea is to demystify power quality analysis. Indeed, both approaches incorporate measures regulated in accordance with the EN 50160 standard. Through better understanding the quality of supply, better decisions can be made regarding energy efficiency and the maintenance of facilities. Measurements and analyses beyond the EN 50160 standard are needed.

There are many types of power quality problems, including voltage lag (voltage values which are outwith the established limits), frequency surge (frequency significantly different to the nominal) and the insertion of harmonics into the power line [10]. Most of the anomalies that take place are in terms of voltage.

The purpose of this study is to provide a comprehensive and unique data set associated with long-term power quality monitoring, which will provide a comprehensive network status, so as to prevent data mismanagement, and achieve more efficient downstream analytics. In order to achieve this goal, a homemade PQ instrument was used, based on signal-acquiring equipment and a server programmed in $Python^{TM}$, rather than a specific PQ analyzer, as in other works, e.g., in [8]. The designed system is described in detail in Section 2.

## 2. Materials and Methods

The instrumentation used for the data collection was deployed in a public building of the University of Cádiz (Spain). The instrumentation's purpose was to provide an end-to-end solution, from data acquisition to data analysis [11]. A single-phase electrical supply voltage signal was taken directly from a traditional wall socket, in order to examine, in situ, the quality of the voltage electrical supply received by the end user.

The voltage data was acquired with a sampling rate of 10 kHz through a chassis and card from the manufacturer NI™. This equipment received the voltage from the electrical

network, and a pulsating signal (1 PPS) from a GPS reference; the latter for temporary stamping of the measures, as explained in [12].

As Gourov N., et al., detailed in [8], this type of acquisition system can be designed to work offline, storing data in memory, or remotely, sending data in real time to the server. For this study, we chose the second option, as the huge amount of data generated would require massive memory storage capacity, which is not necessary for an online system; and because the system was able to work more autonomously, since it was not necessary to disconnect it from the electrical network to transfer the data.

Data was sent from the chassis to a central computer via ethernet. The computer was able to receive and manage data from various sources, allowing multipoint reading. In this study, the computer was running a program developed in Python that carried out the corresponding modifications, and stored the results in the database and on a network-attached storage. The results were thus available for user consultation. The whole measurement & data pipeline architecture is presented in Figure 1.

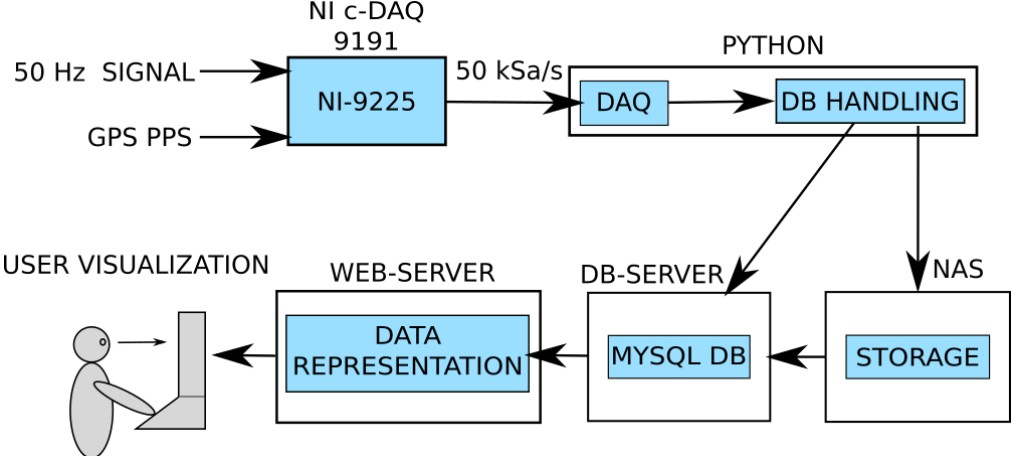

**Figure 1.** Overall measurement & data pipeline architecture. *Grafana* is in the web server.

Frequency measurements were carried out according to two different procedures: a standardized method, used in accordance with the norm [10], storing frequency data every 10 s; and a specific method, in which the frequency and its associated uncertainty were calculated every second, using a method based on *Allan*'s variance. This method, and its benefits over the traditional one, are explained in [13]. In other works—for example, in [4]—a third method was used, based on zero crossings; but as the authors noted, this method was not as reliable, since interharmonics or phase disturbances led to errors in calculations.

In addition to the frequency, other power quality parameters were calculated: variance, kurtosis, skewness, $V_{RMS}$, total harmonic distortion (THD) and a power quality index (PQI) developed in [13]. These calculations were done using three different types of analysis (see Table 1), depending on which computation window was used:

**Table 1.** Analyses performed.

| Analysis Type | Computation Window | Database Update |
|:---:|:---:|:---:|
| 1 | 1 cycle of 50 Hz | 1 s (max, mix, avg values) |
| 2 | 10 cycles of 50 Hz | 1 s (max, mix, avg values) |
| 3 | 2 s | 2 s (value) |

Analysis Type 1 measured waveform time features cycle-by-cycle. Analysis Type 2 measured the same parameters as Type 1, but with a window of 10 cycles, which enabled the calculation of THD. These two analyses returned too many values (one value every 0.02 s for Type 1, and one value every 0.2 s for Type 2); therefore, in order to extract general

information about the grid state, only the minimum, maximum and average values were saved in the database every second, since if all data had been stored there, the queries would have been slower. However, all generated values were saved in files for further consultation if needed.

For Analysis Type 3, results were available every 2 s, and so they were stored in the database directly.

After several tests, it was confirmed that, despite the enormous data flow, the system was working properly. *Python*$^{TM}$ turned out to be so powerful that data processing took only about 15 ms storing $10 \times 10^3$ points, and about 80 ms for $100 \times 10^3$ points, both using a sampling frequency of 10 kHz.

If the system, as it was ultimately designed, had not worked as desired, it would have been necessary to search for alternatives, as was the case in [14], where the developed application used wavelet compression technique for signal data transmission. In that case, before displaying the signal for user visualization, Inverse Discrete Wavelet Transform (IDWT) had to be applied, in order to reconstruct the original signal.

## 3. Data Description and Records

Once the required calculations were done, the next step was to have all data available in a database for further consultation. As explained before, for Analysis Types 1 and Analysis Type 2, the previous step was to store the results in files, that were stored in a Network Attached Storage (NAS) and then averaged, with maximum and minimum values being saved in the database. MATLAB$^{TM}$ files (.mat) were chosen as the primary import source, as they demonstrated a high compression rate, and they could be edited and saved from multiple sources.

Each analysis had its own table on the database, and they were created monthly: Table 2 below is an example of these tables.

**Table 2.** Results from the 2-s scan (Analysis Type 3) for 1 November 2021.

| t | Variance | Skewness | Kurtosis | PQ Index | $V_{RMS}$ (V) | THD | Freq. (Hz) |
|---|---|---|---|---|---|---|---|
| 0:00:01 | 0.506935 | 0.00114295 | −1.50365 | 0.0117251 | 231.59 | 0.0775725 | 49.9511 |
| 0:00:03 | 0.506523 | −0.00129794 | −1.50302 | 0.0108383 | 231.496 | 0.0779946 | 49.9481 |
| 0:00:05 | 0.506305 | −0.00146479 | −1.50178 | 0.00954501 | 231.446 | 0.0807156 | 49.954 |
| 0:00:07 | 0.505414 | −0.00167911 | −1.5017 | 0.00878987 | 231.242 | 0.0705752 | 49.9414 |
| 0:00:09 | 0.506608 | −0.00137201 | −1.50209 | 0.0100683 | 231.515 | 0.085768 | 49.959 |
| 0:00:11 | 0.50713 | −0.000809826 | −1.50355 | 0.0114893 | 231.634 | 0.0804071 | 49.9555 |
| 0:00:13 | 0.506744 | 0.00120363 | −1.50351 | 0.0114553 | 231.546 | 0.0839629 | 49.955 |
| 0:00:15 | 0.506497 | 0.00202925 | −1.50216 | 0.010683 | 231.49 | 0.0821167 | 49.9544 |
| 0:00:17 | 0.50597 | 0.00148796 | −1.50209 | 0.00955296 | 231.369 | 0.0861749 | 49.961 |
| 0:00:19 | 0.505923 | 0.00159949 | −1.50174 | 0.00925834 | 231.359 | 0.0843541 | 49.959 |
| 0:00:21 | 0.506533 | 0.00169597 | −1.50209 | 0.0103187 | 231.498 | 0.0893845 | 49.963 |
| 0:00:23 | 0.506652 | 0.00108132 | −1.50333 | 0.0110593 | 231.525 | 0.0844368 | 49.959 |
| 0:00:25 | 0.50716 | −0.000218186 | −1.50316 | 0.01054 | 231.641 | 0.0905389 | 49.965 |

Data was then available for consultation using *Grafana*$^{TM}$, and also MATLAB$^{TM}$ to represent and assess data. The data was then available for query using Grafana. MATLAB was used only to visualize boxplots, as though it were a punctual complement to the information offered by the trend curves with *Grafana*$^{TM}$. In this way, we were able to eliminate data that was not representative of the main trend of the time series, and to make the interpretation of it more powerful. But the main display engine was *Grafana*$^{TM}$, which is what we show below. The significance of *Grafana*$^{TM}$ as a display option is illustrated below. The main idea was to elicit subjacent trends, tendencies and hidden patterns in the data structures, so as to enable the data operator to easily make decisions and adopt quick measurements using the web server, which would concentrate the variables involved in a control process. Our study precisely employed this method to monitor power quality.

Multiple analyses were conducted and, for each type, the power quality parameters were calculated. Therefore, for example, the RMS was computed every 1 cycle, every 10 cycles, and every 2 s (Figures 2–4):

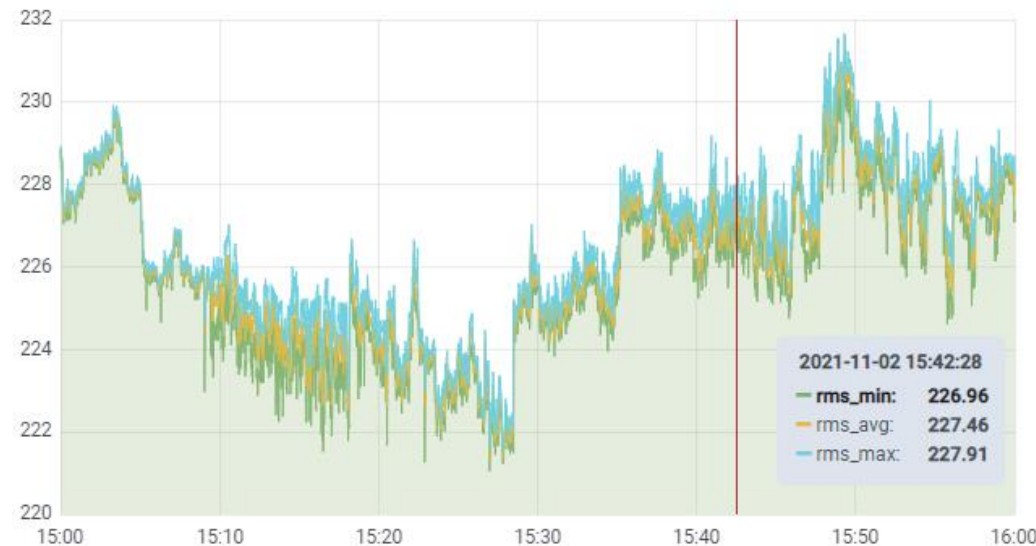

**Figure 2.** RMS values for one cycle analysis (minimum, maximum and average). V$_{RMS}$ vs. hourly time (V vs. hh:mm). The vertical line is an example of cursor reading.

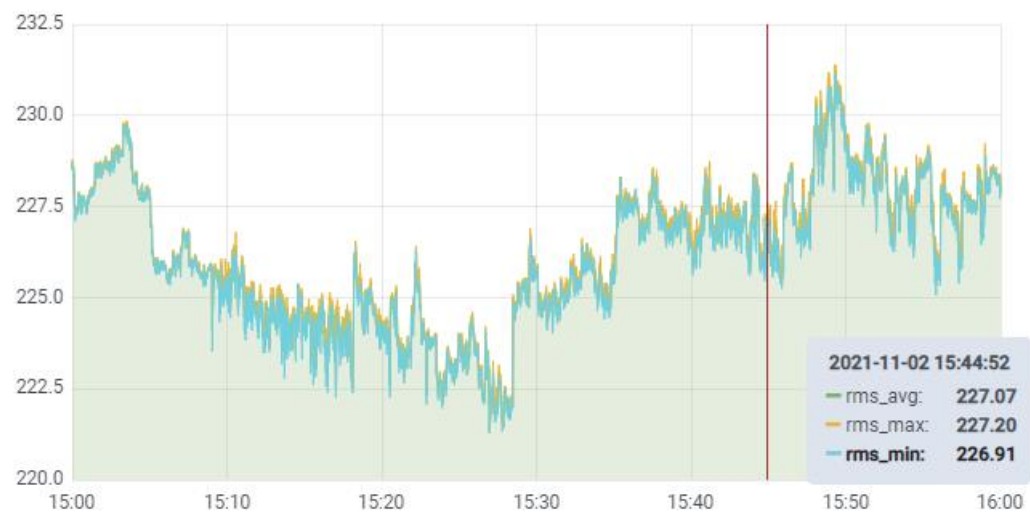

**Figure 3.** RMS values for 10 cycle analysis (minimum, maximum and average). V$_{RMS}$ vs. hourly time (V vs. hh:mm). The vertical line is an example of cursor reading.

The three analyses showed that RMS voltage was oscillating between approximately 222 and 232 V, therefore around 230 V, the nominal value for RMS voltage, according to the UNE 50160 standard.

For frequency it was possible to compare directly the values obtained, following both the method based on *Allan* variance and the traditional method, which follows the [10] standard (see Figure 5).

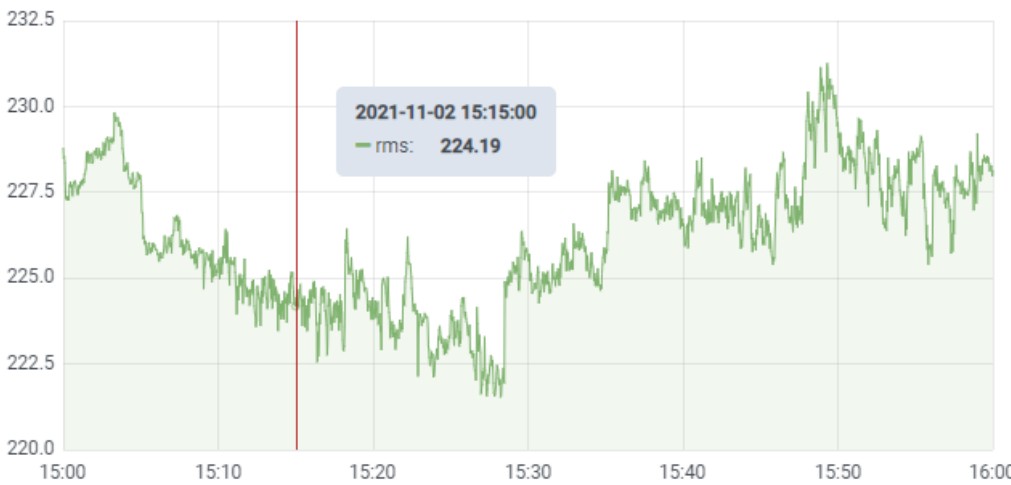

**Figure 4.** RMS values for two second analysis (minimum, maximum and average). V$_{RMS}$ vs. hourly time (V vs. hh:mm). The vertical line is an example of cursor reading.

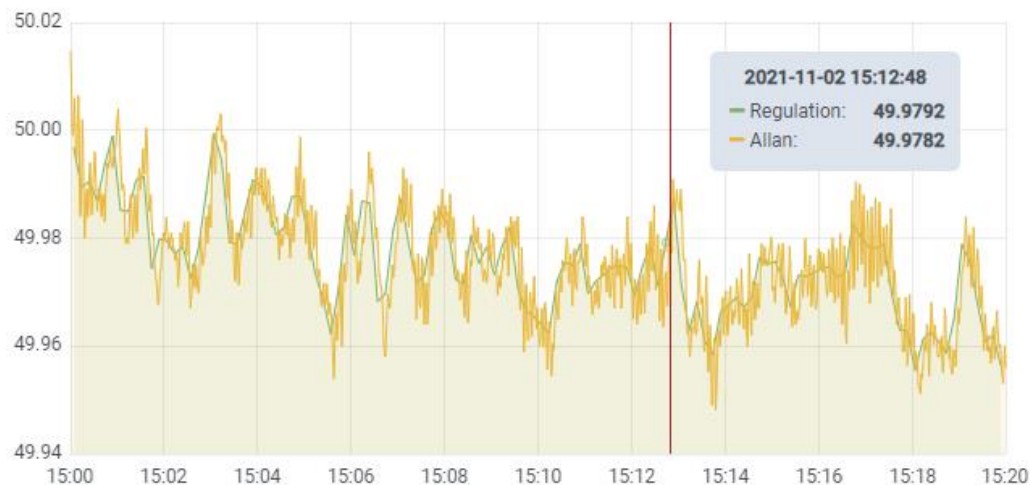

**Figure 5.** Frequency timeline following the standard (green line), and following the algorithm using *Allan* variance (orange line). Frequency vs. hourly time (Hz vs. hh:mm). The vertical line is an example of cursor reading.

It was observed that the tendency of the frequency based on *Allan* variance had a lot of peaks compared to the tendency of the traditional frequency. This was logical as, with the *Allan* variance method, the frequency was calculated every second, while in the traditional method it was calculated every 10 s, therefore the tendency in the latter case looked softer.

By representing the values of the power quality parameters across time, it was possible to detect anomalies, which could be useful for detecting and preventing disturbances in the power network. An example of unusual values for the skewness of the signal can be seen in Figure 6, where two values outside the range of tolerable values were detected:

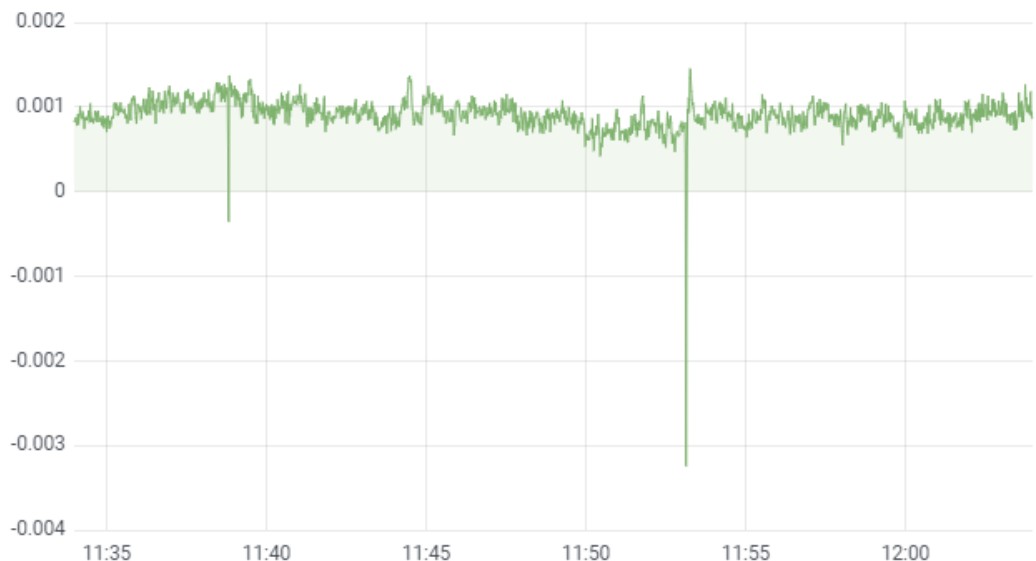

**Figure 6.** Skewness: 1-cycle measurements averaged each second.

These kinds of disturbance can damage sensitive equipment, especially if they persist over time.

As a complement to the former plots, MATLAB$^{\text{TM}}$ offered even more possibilities for data analysis. For example, Figure 7 shows the distribution of the frequency using the tool *Boxplot*:

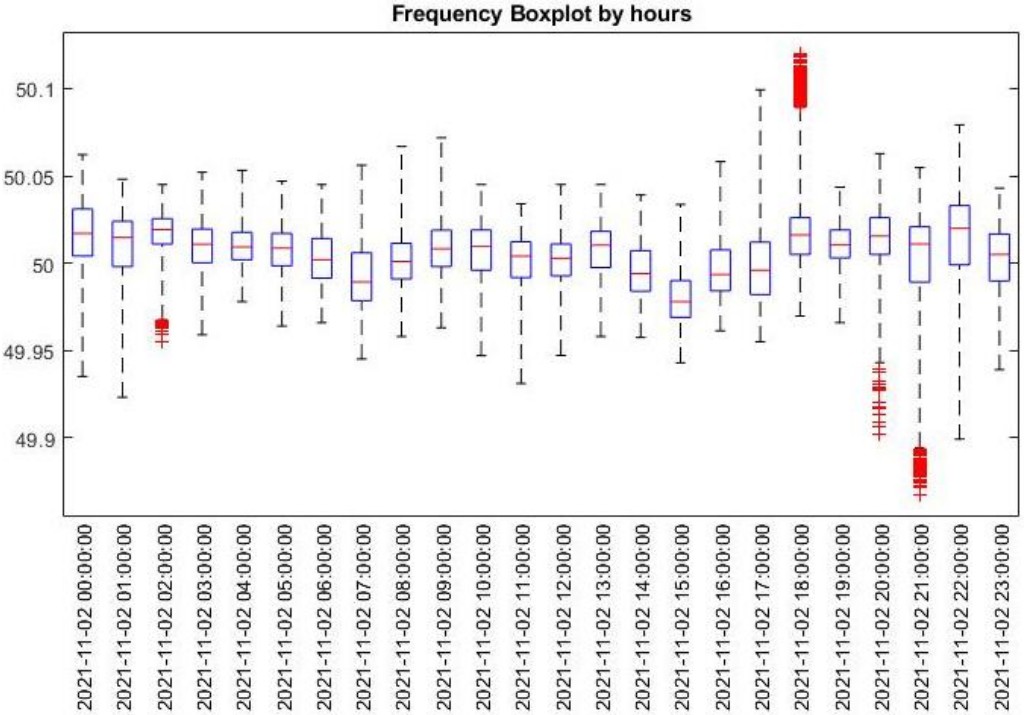

**Figure 7.** Boxplot of frequency values for 1 day, grouped by hours.

This type of graphic classifies data in boxes. There is one box for each hour of the day. The red line inside the box indicates the median of data that belongs to that time interval. The superior and inferior limits of the boxes represent, respectively, the percentiles 75 and 25 of data. The whiskers of the boxes represent those data outwith the percentiles which are not considered atypical, while the red crosses show the values considered anomalous. More details about this type of representation for RMS Voltage can be seen at [15]. It can be

seen observed that data is fluctuating around 50 Hz, the nominal frequency following the UNE 50160 standard.

Another possibility that MATLAB$^{TM}$ offered, over Grafana$^{TM}$, was to front power quality parameters, to find relationships between them. For example, THD vs. frequency has been represented in Figure 8 for data recorded in July 2021. For a better analysis, values considered as outliers were discarded.

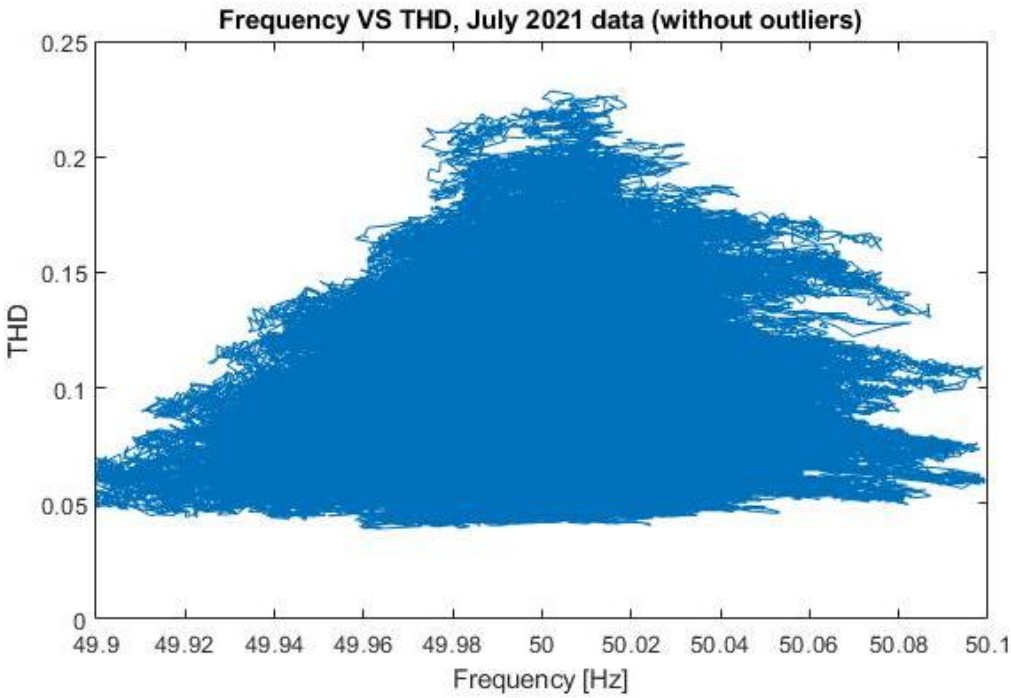

**Figure 8.** THD vs. frequency plot zoomed without outliers.

Figure 8 shows seen that the 'cloud' was approximately symmetric, and that its vertical symmetry axis was situated on 50 Hz. The 'cloud' had a triangular form: low values of THD corresponded to notable variations of frequency, while for a THD of around 0.2, frequency variations were not significant.

## 4. Conclusions

The dataset provides information about the electrical grid, which was variable in time and location, despite its nominal values and tolerances having been established by the norm. Studying the characteristics of the grid made it possible to carry out static and dynamic characterization of the power system, in order to assess the quality of the electricity supply (power quality, PQ).

One of the most critical parameters was frequency, as frequency exerts a decisive influence on the performance of electrical machines. For instance, on filters which require very precise tuning, a significant frequency deviation from the nominal value result is unacceptable. In this study, frequency was monitored by two methods: the traditional one, established by the UNE EN 61000-4-30:2015 norm, in which the measures were taken every 10 s [7], and an alternative method, based on *Allan*'s variance, in which a frequency measure was taken each second, so that it was possible to track the frequency in more detail.

Various statistical parameters were also calculated (e.g., skewness, kurtosis, variance, and THD). They were calculated by different analyses: every cycle of 50 Hz; every 10 cycles; and every 2 s.

It is important to stress that, because data was deposited in the database every 1, 2 or 10 s each day, the monthly tables had a huge number of rows, so that when they were

exported, the resulting files were of a very large size. That is why only 1 month's data was appended to the dataset.

The possible drawbacks of the proposed monitoring system reside in the difficulty of achieving embedded systems conceived to be integrated in the modern industry. This can be the subject of future study.

The data presented in this data descriptor was used to evaluate power quality, which depends largely on distributed generators and non-linear loads, as well as different operating conditions. Due to their high variability, probabilistic studies are inevitable. The study of power quality will help to identify the worst performing areas, and will facilitate the development of appropriate mitigating solutions.

**Author Contributions:** J.-M.S.-F. and M.-J.E.-G. programmed the *Python* codes. J.F.-M. and J.-J.G.-d.-l.R. developed the overall paper organization, and were responsible for experiment design and equipment operation. J.F.-M. generated, analyzed and interpreted graphs. A.A.-P., O.F.-O. and J.-C.P.-S. contributed to interpretation. P.R.-C. did bibliography searching. All authors have read and agreed to the published version of the manuscript.

**Funding:** This research was funded by the Spanish Ministry of Science and Education through the project PID2019-108953RB-C21; it was co-financed by the European Union under the 2014–2020 ERDF Operational Program. Funding for frequency monitoring came from the Andalusian-FEDER project, FEDER-UCA18-108516 (Intelligent Techniques for visualization and data compression of PQ data in the smart grid).

**Institutional Review Board Statement:** Not applicable.

**Informed Consent Statement:** Not applicable.

**Data Availability Statement:** The open data are fully available at https://doi.org/10.7910/DVN/EGI7X1 (accessed on 1 June 2022).

**Acknowledgments:** The authors express their gratitude to the Spanish Ministry of Science and Education for funding the research project PID2019-108953RB-C21, entitled "Strategies for Aggregated Generation of Photo-Voltaic Plants-Energy and Meteorological Data" (SAGPV-EMOD). This work was co-financed by the European Union, under the 2014–2020 ERDF Operational Program and the Regional Government of Andalusia.

**Conflicts of Interest:** The authors declare no conflict of interest.

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
