# Peer review of "Statistical Dataset and Data Acquisition System for Monitoring the Voltage and Frequency of the Electrical Network in an Environment Based on Python and Grafana"

_data, 2022_

Round 1

Reviewer 1 Report

This version is a significant improvement on the previous version.  The contribution and the description of the data pipeline are clear.

What is missing is a statement on the significance of Grafana as a display option.  In Figure 1 Python is marked, but Grafana is not marked. I assume it's in the WEB SERVER.  Also in lines 156/157 you mention that you use both MATLAB and Grafana.  Why not just Grafana? Are the Box/Whisker plots not available? Why not just MATLAB?  Are there performance or access issues? I would like to see a couple of sentences on this point.

Author Response

Dear Reviewer,

Thank you very much for your valuable comments. We have done our best to fulfill your requests. Hereinafter we pass tu ask your questions. Every remarks have been translated into the text by green texts.

1.- Comments and Suggestions for Authors

This version is a significant improvement on the previous version.  The contribution and the description of the data pipeline are clear. 

Response: Thank you. We have worked according to the precise indications.

2.- What is missing is a statement on the significance of Grafana as a display option.  In Figure 1 Python is marked, but Grafana is not marked. I assume it's in the WEB SERVER.  Also in lines 156/157 you mention that you use both MATLAB and Grafana.  Why not just Grafana? Are the Box/Whisker plots not available? Why not just MATLAB?  Are there performance or access issues? I would like to see a couple of sentences on this point.   

Response:

The paper has been read and edited completely in order to get full performance. Specific changes are in red and provide answers to the interesting and meaninful queries.

  • Yes you are right. Grafana is in the web server. We have added this comment in the caption of the figure 1.
  • MATLAB is a complement to Grafana when using boxplots. Perhaps in the future Grafana will incorportate box plots. We have added a consistent sentence to expain this issue.
    • Data is then available for consultation (Figures 2-4) used GrafanaTM and also MATLABTM to represent and assess data. The data is then available for query using GrafanaTM. MATLAB is used only to visualize boxplots, as if it were a punctual complement to the information offered by the trend curves with GrafanaTM. In this way you can eliminate data that are not representative of the main trend of the time series, and make the interpretation of it more powerful. But the main display engine is GrafanaTM, which is what we show below.
  • A statment explaining the significance of Grafana as a visualization tool has been added explaining its potential usage in the context of power quality.
    • The significance of Grafana as a display option is illustrated hereinafter. The main idea is to elicit subjacent trends, tendencies and hidden patterns in the data structures. Consequently, the data operator can easily make a decision and adopt quick measurements using the web server which concentrates the variables involved in a process control. This is precisely what it is pursued in the real-life application involved in this paper, power quality monitoring.
  • MATLAB files have been chosen as sources primary import sources. A comprehensive sentence has been added.
    • MATLABTM files (.mat) have been chosen as primary import sources since it has been demonstrated that a high compression rate and they can be, edited and saved them from multiple sources.

We are at your full disposal for further improvements. Please do not hesitate to contact for further improvements.

Reviewer 2 Report

No more comment

Author Response

Dear Reviewer,

Thank you very much for your valuable comments. We have done our best to fulfill your requests. Hereinafter we pass tu ask your questions. Every remarks have been translated into the text by yellow texts.

1.- Comments and Suggestions for Authors

No more comments. 

Response: Thank you.

We are at your full disposal for further improvements.

This manuscript is a resubmission of an earlier submission. The following is a list of the peer review reports and author responses from that submission.

Round 1

Reviewer 1 Report

The paper is overall quite interesting and serves to promote the data set referenced.

To improve the paper I have the following comments:

  1. The reference [7] for Power Quality is quite weak. There are better references - even if you just bring up the reference to EN61000 [10] to this point in the paper.
  2. Are there not other datasets with similar measurements to compare your results to?
  3. Section 2 is too brief. Here There should be a description of the building where the data was measured, a table with description of the measurements & sampling times, and a figure of the overall measurement & data pipeline architecture.
  4. Line 97 "...it has been proven..." Where? do you have a reference?  Otherwise please justify
  5. Figures 2-10 are poor quality. They need to be at least good enough that the reader can read the numbers on the axes.  
  6. It would help the reader if you also provide a link to a repository where the MATLAB, Grafana, Python and Labview code is provided.
  7. Section 4.1 to improve understanding I suggest diagrams of the 3 programm topologies & a table with the 6 tests.
  8. Given that the times & performance requirements for all tests were well within acceptable limits, I do not understand the value of all the diagrams pp 9-13.  This should be shortened
  9. Why did you not also compare python with matlab? Or compiled matlab? Especially since you use a .mat to store the data.

Overall please compress the paper.

Author Response

Dear Reviewer. Please find attached the responses to your fruitful comments. We have been endeavouring in the enhancement of the paper according to your interesting actions. Please do not hesitate to contact us for further comments. Regards

Reviewer 2 Report

No original information in this article. The power electronic level in this article is very low. The quality of the figures and tables is very poor. The references are mediocre. 

Author Response

Dear reviewer. We have enhanced the paper substantially according to your comments. Please consider that the originality of the paper resides on the basis of a measurement systems that characterises power quality in scenarios where data handling may be critic. Hope to reach your level with this revision. Regards

Reviewer 3 Report

This paper presents set of voltage and current data from a public building and acquired using a hybrid measurement solution combines PythonTM and GrafanaTM. After going through the paper, I found some serious concerns, as listed below, which must be considered before making the final decision:

  1. Authors should add some of the most important quantitative results to the abstract. Focus on the advantages of the proposed method with respect to the obtained results.
  2. Section 1 should be titled “Introduction”
  3. Most of the ideas written were already described in many literatures. The Authors tried to compile it but lack of the enhancement of the interrelation analysis between the references. It is advised that the authors give a deeper analysis on how these ideas become more applicative strategies so that they can contribute to the next step of implementation.
  4. Avoid lumping references such as [2,3,4,5]. Instead summarize the main contribution of each referenced paper in a separate sentence and by including the reference number.
  5. Figures 1 to 10 are not clear, their quality must be enhanced.
  6. Major clarifications and explanations are needed make the contributions of the paper clearly stand out. There exist many works that are focusing on same idea (such as DOI: 1109/IGESSC53124.2021.9618689, DOI: 10.17775/CSEEJPES.2021.01270, https://doi.org/10.3390/en14248304, https://doi.org/10.3390/su132111836 and many more). The literature review should be updated to help readers better understand the subject matter and novelty aspects of this work compared to the recently published works.
  7. More in-depth analysis of the author's contribution of this paper in the introduction section. Moreover, I would like to see more discussion of the literature so that I can clearly identify the article relates to competing ideas.
  8. The authors present only the pros of the approach letting the reviewers to guess the cons. A more critical evaluation of this approach, e.g. in which circumstances it might not work properly, would be welcomed.

Author Response

Dear Reviewer. We have been endeavoring to reach the standard of your fruitful comments. We have detailed the responses in the attached letter in order to make answers more understandable. Please do not hesitate to make further corrections in case needed to enhace the work. Thank you. We are at your disposal.

Round 2

Reviewer 1 Report

Dear Authors, you have not addressed my main concern: That the paper is too long for the contribution it makes.  Similar papers summarize the results in 8 pages including references.  Please revise to be more concise & revise the language.

Reviewer 3 Report

No more comments, however, i hope the authors go again and consider my comments in details. I am not fully satisfied with  the authors response to my comments